# A Radiological Scoring System for Differentiation between Enchondroma and Chondrosarcoma

**DOI:** 10.3390/cancers13143558

**Published:** 2021-07-16

**Authors:** Shinji Miwa, Norio Yamamoto, Katsuhiro Hayashi, Akihiko Takeuchi, Kentaro Igarashi, Kaoru Tada, Hirotaka Yonezawa, Sei Morinaga, Yoshihiro Araki, Yohei Asano, Shiro Saito, Miho Okuda, Junichi Taki, Hiroko Ikeda, Takayuki Nojima, Hiroyuki Tsuchiya

**Affiliations:** 1Department of Orthopedic Surgery, Graduate School of Medical Sciences, Kanazawa University, Kanazawa 920-8640, Japan; norinori@med.kanazawa-u.ac.jp (N.Y.); hayashikatsu830@aol.com (K.H.); a_take@med.kanazawa-u.ac.jp (A.T.); kenken99004@yahoo.co.jp (K.I.); tdkr@med.kanazawa-u.ac.jp (K.T.); yocci1113.papagena.papageno@gmail.com (H.Y.); reddchicke@yahoo.co.jp (S.M.); yaraki87@yahoo.co.jp (Y.A.); you.you.mounin@gmail.com (Y.A.); ssaitomker@gmail.com (S.S.); nojima@kanazawa-med.ac.jp (T.N.); tsuchi@med.kanazawa-u.ac.jp (H.T.); 2Department of Radiology, Graduate School of Medical Sciences, Kanazawa University, Kanazawa 920-8640, Japan; okudamiho193@gmail.com; 3Department of Nuclear Medicine, Graduate School of Medical Sciences, Kanazawa University, Kanazawa 920-8640, Japan; taki@med.kanazawa-u.ac.jp; 4Department of Pathology, Graduate School of Medical Sciences, Kanazawa University, Kanazawa 920-8640, Japan; h-ikeda@med.kanazawa-u.ac.jp

**Keywords:** enchondroma, atypical cartilaginous tumors, chondrosarcoma, differentiation, X-ray, CT, MRI, bone scan, thallium scan, radiological scoring system

## Abstract

**Simple Summary:**

Background: It is challenging to differentiate between enchondromas and atypical cartilaginous tumors (ACTs)/chondrosarcomas. Methods: To evaluate the diagnostic usefulness of radiological findings for differentiation between enchondromas and chondrosarcomas, correlations between various radiological findings and final diagnoses were investigated. Based on the correlations, a scoring system combining these findings was developed. Results: In a cohort of 81 patients, periosteal reaction on X-ray, endosteal scalloping and cortical defect on CT, extraskeletal mass, multilobular lesion, abnormal signal in adjacent tissue on MRI, and increased uptake in bone scan and thallium scan was significantly correlated with final diagnoses. Based on the correlations, a radiological scoring system combining radiological findings was developed. In another cohort of 17 patients, the sensitivity, specificity, and accuracy of the radiological score rates for differentiation between enchondromas and chondrosarcomas were 88%, 89%, and 88%, respectively. Conclusion: Comprehensive assessment combining radiological findings is recommended to differentiate between enchondromas and ACTs/chondrosarcomas.

**Abstract:**

Background: It is challenging to differentiate between enchondromas and atypical cartilaginous tumors (ACTs)/chondrosarcomas. In this study, correlations between radiological findings and final diagnosis were investigated in patients with central cartilaginous tumors. Methods: To evaluate the diagnostic usefulness of radiological findings, correlations between various radiological findings and final diagnoses were investigated in a cohort of 81 patients. Furthermore, a new radiological scoring system was developed by combining radiological findings. Results: Periosteal reaction on X-ray (*p* = 0.025), endosteal scalloping (*p* = 0.010) and cortical defect (*p* = 0.002) on CT, extraskeletal mass (*p* < 0.001), multilobular lesion (*p* < 0.001), abnormal signal in adjacent tissue (*p* = 0.004) on MRI, and increased uptake in bone scan (*p* = 0.002) and thallium scan (*p* = 0.027) was significantly correlated with final diagnoses. Based on the correlations between each radiological finding and postoperative histological diagnosis, a radiological scoring system combining these findings was developed. In another cohort of 17 patients, the sensitivity, specificity, and accuracy of the radiological score rates for differentiation between enchondromas and ACTs/chondrosarcomas were 88%, 89%, and 88%, respectively (*p* = 0.003). Conclusion: Radiological assessment with combined radiological findings is recommended to differentiate between enchondromas and ACT/chondrosarcomas.

## 1. Introduction

Cartilaginous tumors, which are characterized by the formation of a cartilaginous matrix, are one of the most common bone tumors [1]. According to the recent World Health Organization classification, cartilaginous tumors are classified as benign (such as osteochondroma and enchondroma), intermediate (chondromatosis and atypical cartilaginous tumors (ACTs)), and malignant (such as grade I–III chondrosarcoma, dedifferentiated chondrosarcoma, and mesenchymal chondrosarcoma) [2]. In cartilaginous tumors, high-grade area can be focally present due to the morphological heterogeneity of the tumors [3,4,5,6,7,8]. Difficulty in detecting the high-grade area of the tumor may cause sampling failure and under-grading of the tumor [8]. Therefore, diagnostic biopsy is unreliable in evaluating the histological grade in cartilaginous tumors [7].

Cartilaginous tumors can have characteristic radiologic features that allow for differentiation between enchondroma and chondrosarcoma, and recent studies have focused on radiological differentiation between enchondroma and chondrosarcoma [9,10,11,12]. Although the usefulness of several radiological findings has been reported for differentiation between enchondroma and chondrosarcoma [9,10,11,12], the reliability of the findings is unsatisfactory. Therefore, a new method is needed for distinguishing between enchondromas and ACT/chondrosarcomas to aid in the definitive diagnosis of cartilaginous tumors.

Radiological examinations, including X-ray, computed tomography (CT), magnetic resonance imaging (MRI), bone scan, and thallium-201 (^201^Tl) scan, have been widely performed for radiological diagnosis and treatment planning in cartilaginous tumors [13,14,15]. In the management of cartilaginous tumors, the correlation between radiological findings and final diagnoses is thought to be important for predicting histological grades. This study investigated the correlations between various radiological findings and final diagnoses in patients with cartilaginous tumors. Furthermore, a new radiological scoring system combining various radiological findings was developed. The correlation between the radiological scores and histological grade in cartilaginous tumors was evaluated to assess the usefulness of the scoring system.

## 2. Materials and Methods

### 2.1. Development Cohort

This retrospective study included 98 patients with central cartilaginous tumors who underwent biopsy and/or tumor excision at the Department of Orthopedic Surgery of our hospital between October 2003 and December 2020 in our hospital. The patients were divided into a development cohort and validation cohort according to the time of histological diagnoses. Among the patients, 81 patients with central cartilaginous tumors diagnosed between January 2008 and December 2020 were included in the development cohort (Table 1). Inclusion criteria were as follows: central cartilaginous tumors proven pathologically by specimens obtained from biopsy and/or surgery. There were 48 patients with enchondromas, 15 patients with ACT/grade I chondrosarcomas, 12 patients with grade II chondrosarcomas, 1 patient with grade III chondrosarcoma, 4 patients with dedifferentiated chondrosarcoma, and 1 patient with mesenchymal chondrosarcoma. There were 42 men and 39 women, with a mean age of 42.8 years (range, 4–89 years). In all cases, tumor specimens were assessed histologically, and final diagnoses were made by pathologists. Exclusion criteria were as follows: recurrent tumors, metastatic tumors, and tumors without histological diagnoses. This study protocol was approved by the Medical Ethics Committee of Kanazawa University (IRB number: 2020-125). Informed consent was obtained using the opt-out method.

### 2.2. Correlation of Radiological Findings and Diagnoses

All the 81 cases were assessed by X-ray, 57 cases were assessed by CT, 70 cases were assessed by MRI, 36 cases were assessed by bone scan, and 29 cases were assessed by ^201^Tl scan. In the X-ray photographs, cortical expansion, periosteal reaction, and the presence of pathologic fractures were assessed [16]. On CT, endosteal scalloping and cortical defects were assessed [17]. Endosteal scalloping was defined as cortical thinning of more than two-thirds of the cortical thickness [9]. On MRI, the presence of an extraskeletal mass, multilobular lesion, and abnormal signal in the adjacent bone marrow and soft tissue were assessed [9,18]. In bone scans, increased uptake of ^99m^technetium-methylene diphosphonate (^99m^Tc-MDP) was defined as that with a greater uptake than the anterior iliac crest [13,19]. Bone scans were evaluated among patients without pathologic fracture because fractures usually cause increased uptake of ^99m^Tc-MDP. In the ^201^Tl scan, increased uptake was defined as the increased accumulation of ^201^Tl compared with the contralateral normal area or the adjacent area [14]. To evaluate the usefulness of radiological examinations, sensitivity, specificity, negative predictive value (NPV), positive predictive value (PPV), accuracy, and kappa (K) values for differentiation between enchondromas and ACT/chondrosarcomas were investigated.

### 2.3. A Radiological Scoring System for Differentiation between Enchondroma and Chondrosarcoma

To differentiate between enchondroma and ACT/chondrosarcoma and to predict the histological grade of cartilaginous tumors, an overall radiological scoring system was developed by combining radiological findings from X-ray, CT, MRI, bone scan, and ^201^Tl scan. Radiological findings that were significantly correlated with final diagnoses were included in the scoring system. Cohen’s kappa coefficients (K values) were used to evaluate the concordance between radiological findings and histological diagnoses. Among the radiological findings, three points were assigned to the findings with a K value ≥ 0.5, 2 points were assigned to the findings with a K value < 0.5 and ≥0.28, and 1 point was assigned to findings with a K value < 0.28. The sensitivity, specificity, NPV, PPV, accuracy, and K value of the radiological scores for differentiation between enchondromas and ACTs/chondrosarcomas were evaluated. Furthermore, radiological score rates were compared among enchondromas, ACT/grade I chondrosarcomas, grade II chondrosarcomas, and grade III/dedifferentiated/mesenchymal chondrosarcoma.

### 2.4. Validation Cohort

Seventeen patients with central cartilaginous tumors diagnosed between October 2003 and December 2007 were included in the validation cohort (Table 1). In the validation cohort, the inclusion and exclusion criteria were the same as those for the development cohort. To assess the reproducibility of the radiological scores for differentiation between enchondroma and ACT/chondrosarcoma, the sensitivity, specificity, NPV, PPV, accuracy, an K value of the radiological scores were evaluated.

### 2.5. Statistical Analysis

To assess the correlation between radiological findings (cortical expansion, periosteal reaction, pathologic fracture on X-ray, endosteal scalloping and cortical defects on CT, extraskeletal mass, multilobular lesion, and abnormal signal in adjacent bone marrow and soft tissue on MRI, increased uptake on bone scan, and increased uptake on ^201^Tl scan) and histological diagnoses of chondrosarcoma, Fisher’s exact test was performed. Statistical significance was set at *p* < 0.05. To assess the usefulness of radiological findings and radiological score, sensitivity, specificity, NPV, PPV, accuracy, and Κ value were evaluated. The optimal cutoff level of the radiological score was identified as each index value minimizing the number of false results by receiver operator characteristic (ROC) curve analysis. The radiological scores of enchondroma, ACTs/grade I chondrosarcoma, grade II chondrosarcoma, and grade III/dedifferentiated/mesenchymal chondrosarcoma were evaluated by analysis of variance (ANOVA) followed by Bonferroni’s test. EZR (Saitama Medical Center, Jichi Medical University, Saitama, Japan) was used for the statistical analyses.

## 3. Results

### 3.1. Correlation between Radiological Findings and Histological Diagnoses

Among the radiological findings, the presence of periosteal reaction on radiography (*p* = 0.025); endosteal scalloping (*p* = 0.010) and cortical defect (*p* = 0.002) on CT; extraskeletal mass (*p* < 0.001), multilobular lesion (*p* < 0.001), and abnormal signal intensity in adjacent bone marrow and soft tissue (*p* = 0.004) on MRI; and increased uptake on bone scan (*p* = 0.002) and ^201^Tl scan (*p* = 0.027) were significantly correlated with the diagnosis of ACT/chondrosarcoma (Table 2). To diagnose ACT/chondrosarcoma, sensitivity, specificity, and accuracy were 12.1%, 100.0%, and 64.2% for periosteal reaction on X-ray (K = 0.141); 96.7%, 29.6%, and 64.9% for endosteal scalloping on CT (K = 0.272); 86.7%, 55.6%, and 71.9% for cortical defect on CT (K = 0.429); 61.3%, 100.0%, and 82.9% for extraskeletal mass on MRI (K = 0.638); 80.6%, 92.3%, and 87.1% for multilobular lesion on MRI (K = 0.737); 32.3%, 94.9%, and 67.1% for abnormal signal in adjacent bone marrow and soft tissue on MRI (K = 0.289); 100.0%, 41.7%, and 80.6% for increased uptake on bone scan (K = 0.488); and 45.0%, 100.0%, and 62.1% for increased uptake on ^201^Tl scan (K = 0.337), respectively (Table 3).

### 3.2. A Radiological Scoring System Combining Radiological Findings for Differentiation between Enchondroma and ACTs/Chondrosarcoma

To establish a comprehensive method for differentiation between enchondromas and ACTs/chondrosarcomas, an overall radiological scoring system combining findings from X-ray, CT, MRI, bone scan, and ^201^Tl scan was developed. Based on the K values of the radiological findings for differentiation between enchondroma and ACTs/chondrosarcoma, 3 points were attached to the extraskeletal mass on MRI and multilobular lesion on MRI; 2 points were attached to cortical defect on CT, and abnormal signal in the adjacent bone marrow and soft tissue on MRI, increased uptake on bone scan, and increased uptake on ^201^Tl scan; and 1 point was attached to the periosteal reaction in the X-ray and endosteal scalloping on CT (Table 4). The overall radiological score rate for cartilaginous tumors was determined as follows:Overall radiological score rate = total radiological scores/full marks × 100 (%).

In patients lacking radiological images, the overall radiological score rates were calculated only based on the findings available. The cutoff value of 31.3% was calculated by ROC curve analysis for differentiation between enchondroma and ACT/chondrosarcoma, as each index value minimized the number of false results (Figure 1). In the development cohort of 81 patients, the sensitivity, specificity, NPV, PPV, and accuracy of the overall radiological score rate for differentiation between enchondromas and ACTs/chondrosarcomas were 100.0%, 85.4%, 100.0%, 82.5%, and 91.4%, respectively (K = 0.827; Table 5).

Among the cartilaginous tumors, the overall radiological score rates were 11.6 ± 16.2% in patients with enchondroma, 62.1 ± 22.7% in patients with ACT/grade I chondrosarcoma, 64.7 ± 21.2% in patients with grade II chondrosarcoma, and 84.3 ± 24.3% in patients with grade III/dedifferentiated/mesenchymal chondrosarcoma, respectively (Figure 2). Figure 3, Figure 4 and Figure 5 show images of enchondroma (Figure 3), ACT/grade I chondrosarcoma (Figure 4), and high-grade chondrosarcoma (Figure 5), respectively, from our study.

In the validation cohort, overall radiological scores ≥31.3% were significantly correlated with the diagnosis of ACT/chondrosarcoma (*p* = 0.003). In the cohort, the sensitivity, specificity, NPV, PPV, and accuracy of the overall radiological score rate for differentiation between enchondromas and ACTs/chondrosarcomas were 87.5%, 88.9%, 88.9%, 87.5%, and 88.2%, respectively (K = 0.764; Table 5).

## 4. Discussion

In the treatment of cartilaginous tumors, surgical procedures are determined by the histological grade of the tumor, obtained by biopsy [20]. Unlike other types of bone tumors, a definitive diagnosis of cartilaginous tumors by biopsy is thought to be difficult [7]. Cartilaginous tumors commonly have morphological heterogeneity and can have focal high-grade areas [21]. Consequently, there are risks of sampling failure of high-grade tumor tissue in a predominantly benign or low-grade lesion, and of under- or over-grading the tumors [22,23]. Compared with other types of bone tumors, high rates of discrepancies between biopsy findings and final diagnoses were reported in cartilaginous tumors [3,5,7,24]. In previous studies, 14–57% of patients with cartilaginous tumors had discrepancies in the histological grade between biopsy results and final diagnosis [5,7,25,26]. Especially, histological distinction of ACT/grade I chondrosarcoma from enchondroma is difficult in many cases for pathologists because of their similar cytology, cellularity, and cartilaginous matrix [9,27]. In cases with discrepancies between biopsy results and final diagnosis, additional tumor excision or radiation therapy may be required. To select the appropriate surgical treatment, preoperative assessments with high accuracy for the prediction of histological grades are required in cartilaginous tumors.

Previous studies have reported the usefulness of radiological findings in X-ray, CT, MRI, and nuclear medicine [9,14,15,16,20,28,29,30,31]. Although these radiological examinations are commonly used for making this distinction [15,32], there is no standard evaluation method of the radiological examinations. In the present study, the periosteal reaction on X-ray, and endosteal scalloping and cortical defect on CT were significantly correlated with the final diagnosis. In previous reports, periosteal reaction and cortical thickening were more frequently observed in chondrosarcomas [13,33], although the usefulness of X-rays is controversial because of the difficulty of objective assessment of the X-ray images [16,34]. Geirnaerdt et al. reported that ill-defined margins were observed in 67% of chondrosarcomas and 37% of enchondromas (*p* = 0.004), and that multilobulated lesions were observed in 72% of chondrosarcomas and 43% of enchondromas (*p* = 0.009) [16]. Murphey et al. reported that endosteal scalloping depth was seen in 90% of chondrosarcomas and 10% of enchondromas, and that cortical defects were observed in 88% of chondrosarcomas and 8% of enchondromas [13].

In this study, overall radiological scores were highly correlated with the chondrosarcoma grades. There are some studies on the differentiation between enchondromas and chondrosarcomas using a combination of clinical features [19]. Ferrer-Santacreu et al. proposed a scoring system combining pain on palpation, cortical involvement in CT or MRI, and increased uptake in the bone scan [19]. In their report, the sensitivity and specificity for differentiation between enchondromas and low-grade chondrosarcomas were 74% and 94%, respectively [19]. In another study, Parlier-Cuau et al. proposed a diagnostic strategy for the management of cartilaginous tumors [33]. In the report, radiological findings were divided based on “active” findings and “aggressive” findings. In the report, pain related to the lesion, endosteal scalloping >2/3 of the cortical thickness, the extent of endosteal scalloping along >2/3 of the lesion length, cortical thickness or hyperostosis, cortical remodeling with enlargement of the diameter of the medullary cavity, delayed bone scintigram showing an intense uptake greater than that of the anterior iliac crest in the absence of fracture, and early and exponential enhancement on dynamic gadolinium-enhanced MR sequences were included in “active” findings. However, pathologic fracture arising with minimal trauma, periosteal reaction, moth-eaten or permeative osteolysis, and the presence of a soft tissue mass were included in “aggressive” findings. The authors recommended biopsy for cases with an aggressive finding or two or more active findings. Although radiological examinations have errors, this study demonstrated that comprehensive assessment of the examinations could reduce the errors and showed high accuracy for differentiation between enchondromas and chondrosarcomas. This revealed a high correlation between radiological scores and histological grades of cartilaginous tumors. Although this scoring system accepts assessments by a few examinations, assessments of three or more radiological examinations, including CT, MRI, and nuclear medicine, are recommended to reduce diagnostic errors.

This study had several limitations, the most important being the heterogenous tumor locations and the small number of patients. In the latest World Health Organization classification, tumors in the extremities are termed central ACTs, whereas those in the trunk are termed grade I chondrosarcomas [2]. Therefore, tumor location is an important factor in the determination of the histological grade of cartilaginous tumors. Further studies including a large number of patients with cartilaginous tumors in only the extremities or only the trunk are required to improve the comprehensive method of radiological assessment.

## 5. Conclusions

The present study showed that periosteal reaction on X-ray; endosteal scalloping and cortical defect on CT; multilobular lesion, extraskeletal mass, abnormal signal in adjacent bone marrow, and soft tissue on MRI; and increased uptake on bone scan and ^201^Tl scan are useful in differentiating between enchondroma and chondrosarcoma. Furthermore, the overall radiological scoring system combining radiological findings is highly correlated with the histological grade of cartilaginous tumors. Comprehensive assessment combining radiological findings is recommended to predict the histological grades of cartilaginous tumors.

## Figures and Tables

**Figure 1 cancers-13-03558-f001:**
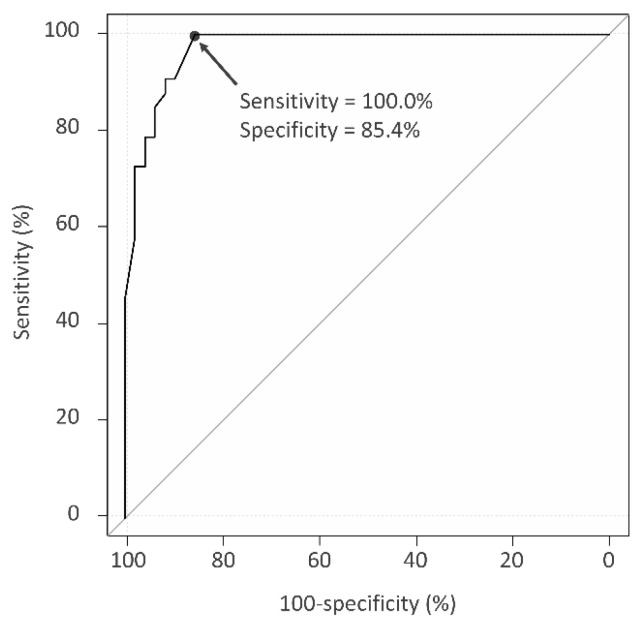
Receiver operator characteristic (ROC) curve for the prediction of ACT/chondrosarcoma with overall radiological score rates. The cutoff value determined by the ROC curve analysis was 31.3%.

**Figure 2 cancers-13-03558-f002:**
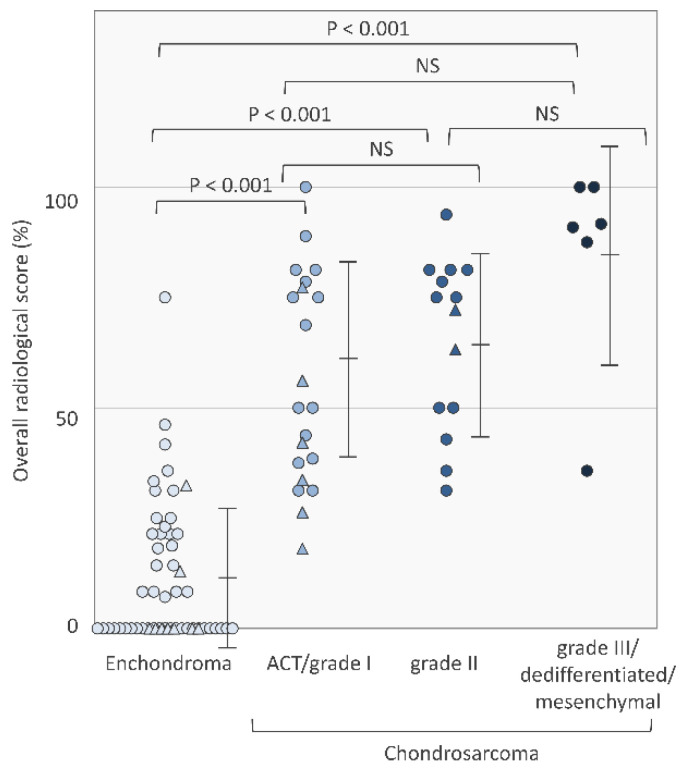
Overall radiological score rates in enchondroma, ACT/grade I chondrosarcoma, grade II chondrosarcoma, and grade III/dedifferentiated/mesenchymal chondrosarcomas. Overall radiological score rates were 11.6 ± 16.2% in patients with enchondroma, 62.1 ± 22.7% in patients with ACT/grade I chondrosarcoma, 64.7 ± 21.2% in patients with grade II chondrosarcoma, and 84.3 ± 24.3% in patients with grade III/dedifferentiated/mesenchymal chondrosarcoma, respectively. Data points with a circle shape are the development cohort and data points with a triangle shape are the validation cohort. Values are expressed as means ± SD. NS: not significant.

**Figure 3 cancers-13-03558-f003:**
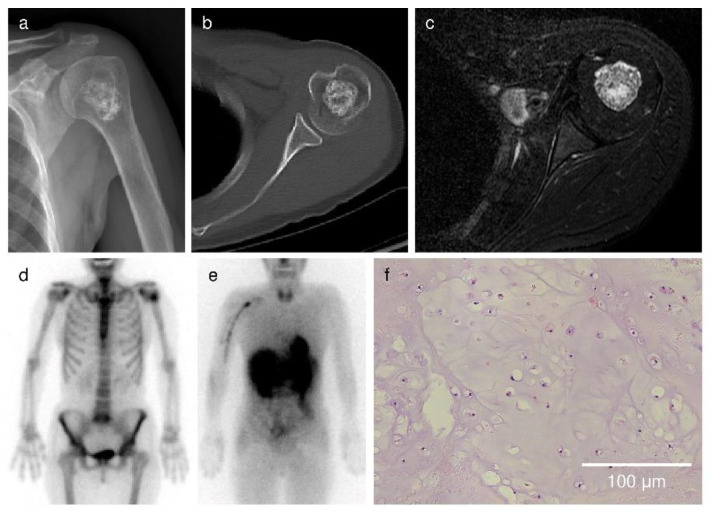
Case 1. A 58-year-old woman presenting with shoulder pain. (**a**) Radiograph showed no periosteal reaction in the proximal humerus. (**b**) There was no endosteal scalloping or cortical defect on CT. (**c**) MRI showed no extraskeletal mass, multilobular lesion, or abnormal signal in adjacent tissue. (**d**) The bone scan demonstrates uptake of ^99m^Tc-MDP in the proximal humerus, but the uptake was not greater than uptake in the anterior iliac crest. (**e**) ^201^Tl scan showed no increased uptake in the proximal humerus. Based on the results, the overall radiological score rate was calculated as follows: [0 points (X-ray) + 0 points (CT) + 0 points (MRI) + 0 points (bone scan) + 0 points (^201^Tl scan)]/16 points (full marks) × 100 = 0%. (**f**) The microscopic image shows hypocellular tumor cells with an abundance of hyaline cartilage matrix. No cytological atypia or mitosis was observed. The final histological diagnosis was enchondroma.

**Figure 4 cancers-13-03558-f004:**
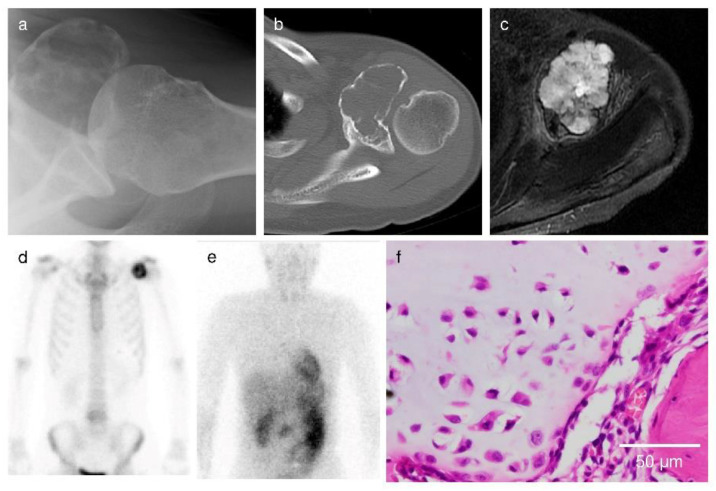
Case 2. A 48-year-old woman presenting shoulder pain. (**a**) The radiograph showed no periosteal reaction. (**b**) CT showed endosteal scalloping and cortical defect. (**c**) MRI showed multilobular lesion, whereas no extraskeletal mass and abnormal signal in adjacent tissue was seen. (**d**) Bone scan showed increased uptake greater than the anterior iliac crest. (**e**) ^201^Tl scan showed no increased uptake in the proximal humerus. Based on the findings, the overall radiological score rate was calculated as follows: [0 points (X-ray) + 3 points (CT) + 3 points (MRI) + 2 points (bone scan) + 0 points (^201^Tl scan)]/16 points (full marks) = 50.0%. (**f**) The microscopic image showed moderate cellularity, with cells embedded in hyaline matrix. A closed chromatin pattern and inconspicuous nucleoli were shown. The final histological diagnosis was grade I chondrosarcoma.

**Figure 5 cancers-13-03558-f005:**
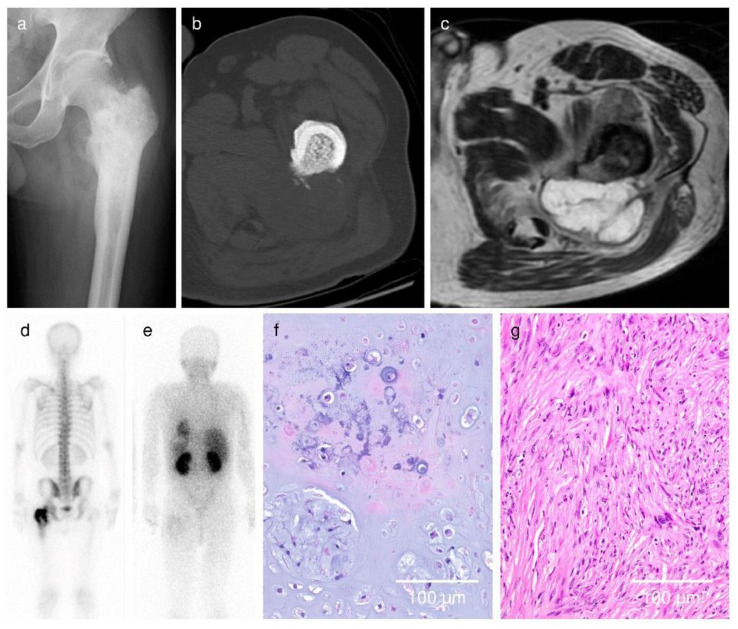
Case 3. A 58-year-old man showed a periosteal reaction on X-ray (**a**); endosteal scalloping and cortical defect on CT (**b**); and extraskeletal mass, multilobular mass, and abnormal signal intensity in adjacent soft tissue and bone marrow on MRI (**c**). Increased uptake greater than the anterior iliac crest on the bone scan and ^201^Tl scan were also seen (**d**,**e**). Based on the findings, the overall radiological score rate was calculated as follows: [1 point (X-ray) + 3 points (CT) + 8 points (MRI) + 2 points (bone scan) + 2 points (^201^Tl scan)]/16 points (full marks) = 100% (**f**,**g**) The microscopic image showed bimorphic appearance of grade 1–2 chondrosarcoma and fibrosarcoma components. The final histological diagnosis of the tumor was dedifferentiated chondrosarcoma.

**Table 1 cancers-13-03558-t001:** Characteristics of study patients.

Characteristics	Development Cohort	Validation Cohort
**Age**	42.8 (4–89)	46.2 (19–74)
**Gender**	42 men/39 women	11 men/6 women
**Diagnoses**		
Enchondroma	48	9
ACT/Grade I chondrosarcoma	15	6
Grade II chondrosarcoma	12	2
Grade III chondrosarcoma	1	0
Dedifferentiated chondrosarcoma	4	0
Mesenchymal chondrosarcoma	1	0
**Tumor location**		
Finger/hand	25	6
Femur	16	4
Humerus	11	0
Pelvis	10	2
Tibia	3	2
Rib	3	2
Spine	4	0
Foot	4	0
Fibula	2	1
Sternum	1	0
Scapula	2	0
**Pathologic fracture**	14	1

ACT: atypical cartilaginous tumors.

**Table 2 cancers-13-03558-t002:** Correlations between radiological findings and final diagnoses.

	Enchondroma	ACT/Chondrosarcoma	OR	95% CI	*p* Value
**X-ray**					
Cortical expansion	23/48	12/33	0.625	0.225–1.682	0.365
Periosteal reaction	0/48	4/33	Inf	1.007–Inf	0.025
Pathologic fracture	11/48	3/33	0.341	0.056–1.450	0.140
**CT**					
Endosteal scalloping	19/27	29/30	11.741	1.389–558.304	0.010
Cortical defect	12/27	26/30	7.794	1.948–39.413	0.002
**MRI**					
Extraskeletal mass	0/39	19/31	Inf	12.161–Inf	<0.001
Multilobular mass	3/39	25/31	45.443	9.889–309.128	<0.001
Abnormal signal in adjacent bone marrow and soft tissue	2/39	10/31	8.540	1.603–87.562	0.004
**Bone scan**					
^99m^Tc-MDP uptake	7/12	24/24	Inf	2.357–Inf	0.002
**Tl scan**					
^201^Tl uptake	0/9	9/20	Inf	1.160–Inf	0.027

**Table 3 cancers-13-03558-t003:** The predictive powers of each radiological findings compared with histological diagnoses.

	Sensitivity	Specificity	NPV	PPV	Accuracy	Kappa Value
**X-ray**						
Periosteal reaction	12.1	100.0	62.3	100.0	64.2	0.141
**CT**						
Endosteal scalloping	96.7	29.6	88.9	60.4	64.9	0.272
Cortical defect	86.7	55.6	78.9	68.4	71.9	0.429
**MRI**						
Extraskeletal mass	61.3	100.0	76.5	100.0	82.9	0.638
Multilobular lesion	80.6	92.3	85.7	89.3	87.1	0.737
Abnormal signal in adjacent bone marrow and/or soft tissue	32.3	94.9	63.8	83.3	67.1	0.289
**Nuclear medicine**						
Bone scan	100.0	41.7	100.0	77.4	80.6	0.488
^201^Tl scan	45.0	100.0	45.0	100.0	62.1	0.337

**Table 4 cancers-13-03558-t004:** Overall radiological scoring system for differentiation between enchondroma and ACT/chondrosarcoma.

	Findings	Points
**X-ray**	Periosteal reaction	1
**CT**	Endosteal scalloping	1
	Cortical defect	2
**MRI**	Extraskeletal mass	3
	Multilobular lesion	3
	The abnormal signal in the adjacent bone marrow and/or soft tissue	2
**Bone scan**	Increased uptake	2
**^201^Tl scan**	Increased uptake	2

**Table 5 cancers-13-03558-t005:** The predictive powers of the overall radiological score compared with histological diagnoses in the developing cohort and validation cohort.

	Sensitivity	Specificity	NPV	PPV	Accuracy	Kappa Value
Developing cohort	100.0	85.4	100.0	82.5	91.4	0.827
Validation cohort	87.5	88.9	88.9	87.5	88.2	0.764

## Data Availability

The datasets supporting the conclusion of this article are included within the article. The underlying datasets are available from the corresponding author upon request.

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
