# Peer review of "A Radiological Scoring System for Differentiation between Enchondroma and Chondrosarcoma"

_cancers, 2021, doi:10.3390/cancers13143558_

Round 1

Reviewer 1 Report

The authors have adressed my recommendations for revision of their manuscript. The "validation cohort" only comprises 17 patients and especially high grade sarcomas are under-representated, however, the results obtained in this cohort support the findings in the "development cohort". As reviewer #2 emphasized, the sample number is limited, but was increased by roughly 30%.

Minor revision:

I suggest to distinguish the two cohorts in Fig. 2 using dots and triangles or any other way.

The limitations of the study are obvious, but were discussed in the discussion. I now support the acceptance of the manuscript.

Author Response

Comments and Suggestions for Authors

The authors have adressed my recommendations for revision of their manuscript. The "validation cohort" only comprises 17 patients and especially high grade sarcomas are under-representated, however, the results obtained in this cohort support the findings in the "development cohort". As reviewer #2 emphasized, the sample number is limited, but was increased by roughly 30%.

Minor revision:

I suggest to distinguish the two cohorts in Fig. 2 using dots and triangles or any other way.

The limitations of the study are obvious, but were discussed in the discussion. I now support the acceptance of the manuscript.

Response:

Thank you for your comments and recommendation.

In Figure 2, two cohorts were distinguished using dots and triangles, and the following sentence was added to the figure legend.

Data points with circle shape are development cohort and data points with triangle shape are validation cohort.

Reviewer 2 Report

Manuscript entitled "A radiological scoring system for differentiation between enchondroma and chondrosarcoma".

This work is still limited by less relevant and not useful clinically. In the other hand, it also lacks novelty.

Author Response

Comments and Suggestions for Authors

Manuscript entitled "A radiological scoring system for differentiation between enchondroma and chondrosarcoma".

This work is still limited by less relevant and not useful clinically. In the other hand, it also lacks novelty.

Response:

Thank you for your comments.

Because the accuracy of histological diagnosis of cartilaginous tumors remains unsatisfactory, we believe that our data will support the surgical planning in patients with discrepancy between radiological findings and biopsy diagnosis.

The manuscript was revised and improved according to the reviewer’s comments.

If there are any further issues that need to be resolved to improve our manuscript, please let us know.

Reviewer 3 Report

The manuscript entitled “A radiological scoring system for differentiation between en-2 chondroma and chondrosarcoma” is an interesting study focused on the development of a new score to distinguish between enchondroma and chondrosarcoma.

My comments are as follow:

1) In the introduction, the authors report “According to the recent World Health Organization classification, cartilaginous tumors are classified as benign (such as osteochondroma and enchondroma), intermediate (chondromatosis and atypical cartilaginous tumors), and malignant (such as grade I–III chondrosarcoma and  dedifferentiated chondrosarcoma)” and in the discussion “latest World Health Organization classification, tumors in the extremities are termed central atypical cartilaginous tumors, whereas those in the trunk are termed grade I chondrosarcomas”. In the description made by the authors in the introduction, it seems that atypical cartilagineous tumour and chodrosarcoma grade 1 are two different type of chondrosarcomas. However, WHO classification (5th edition) define chondrosarcomas grade 1 or atypical cartilaginous tumour / (ACT/CS1), specifying that “Tumours in the appendicular skeleton (long and short tubular bones) are termed ACTs, whereas tumours of the axial skeleton (flat bones, including the pelvis, scapula, and skull base) should be called CS1s.” Thus, I suggest to use “ACT/CS1” instead of “grade I chondorsarcoma” to define these patients. Moreover, the authors did not mention mesenchymal chondrosarcomas.

2) In the introduction, the authors state that it is challenging to differentiate between enchondroma and chondrosarcoma. This point needs to be better explained.

3) Lines 59-61: the authors should add at least a reference.

4) Table 1 is confusing. In the paragraph “2.1. Development cohort”, the authors describe the 81 patients enrolled citing table 1. However, the reader is confused to see also “a validation cohort” that it is not describe in this part of the manuscript. The authors should consider to modify the table or move it at the beginning of the results.

5) Table 4 should be placed before figure 1.

6) Legends of Figures 3,4 and 5 should be improved. The authors should add letters on the figures describing x-ray, RMI, CT and histological images. Magnification/scale bar of histological images is missing.  

7) The authors should better organize the order of figures/tables following the text of the manuscript. Figures and tables should be placed near where they are cited. For example, table 5 should be moved after figure 5.

Minor comments:

It is not clear to me why there are some words in red throughout the manuscript.

Author Response

Comments and Suggestions for Authors

The manuscript entitled “A radiological scoring system for differentiation between enchondroma and chondrosarcoma” is an interesting study focused on the development of a new score to distinguish between enchondroma and chondrosarcoma.

My comments are as follow:

1) In the introduction, the authors report “According to the recent World Health Organization classification, cartilaginous tumors are classified as benign (such as osteochondroma and enchondroma), intermediate (chondromatosis and atypical cartilaginous tumors), and malignant (such as grade I–III chondrosarcoma and  dedifferentiated chondrosarcoma)” and in the discussion “latest World Health Organization classification, tumors in the extremities are termed central atypical cartilaginous tumors, whereas those in the trunk are termed grade I chondrosarcomas”. In the description made by the authors in the introduction, it seems that atypical cartilagineous tumour and chodrosarcoma grade 1 are two different type of chondrosarcomas. However, WHO classification (5th edition) define chondrosarcomas grade 1 or atypical cartilaginous tumour / (ACT/CS1), specifying that “Tumours in the appendicular skeleton (long and short tubular bones) are termed ACTs, whereas tumours of the axial skeleton (flat bones, including the pelvis, scapula, and skull base) should be called CS1s.” Thus, I suggest to use “ACT/CS1” instead of “grade I chondorsarcoma” to define these patients. Moreover, the authors did not mention mesenchymal chondrosarcomas.

Response:

Thank you for your pointing this out. The term of “atypical cartilaginous tumor” was added to this manuscript. According to the latest WHO classification (5th edition), low-grade cartilaginous tumors in the appendicular skeleton (long and short tubular bones) were classified into ACTs, whereas those in axial skeleton (flat bones, including the pelvis, scapula, and skull base) were classified into grade I chondrosarcoma. Therefore, the term of “grade I chondrosarcoma” was replaced by “Atypical cartilaginous tumor (ACT)/grade I chondrosarcoma”.

Mesenchymal chondrosarcoma was added to the Introduction section.

2) In the introduction, the authors state that it is challenging to differentiate between enchondroma and chondrosarcoma. This point needs to be better explained.

Response:

Thank you for your suggestion.

In the introduction, the sentences were revised as follows:

In cartilaginous tumors, high-grade area can be focally present due to the morphological heterogenity of the tumors. Difficulty in detecting the high-grade area of the tumor may cause sampling failure and under-grading of the tumor. Therefore, diagnostic biopsy is unreliable in evaluating the histological grade in cartilaginous tumors.

Cartilaginous tumors can have characteristic radiologic features that allow for differentiation between enchondroma and chondrosarcoma, and recent studies have focused on radiological differentiation between enchondroma and chondrosarcoma. Although usefulness of several radiological findings has been reported for differentiation between enchondroma and chondrosarcoma, the reliability of the findings are unsatisfactory.

References:

  1. Roitman, P. D.; Farfalli, G. L.;  Ayerza, M. A.;  Muscolo, D. L.;  Milano, F. E.; Aponte-Tinao, L. A., Is Needle Biopsy Clinically Useful in Preoperative Grading of Central Chondrosarcoma of the Pelvis and Long Bones? Clin Orthop Relat Res 2017, 475 (3), 808-814.
  2. Laitinen, M. K.; Stevenson, J. D.;  Parry, M. C.;  Sumathi, V.;  Grimer, R. J.; Jeys, L. M., The role of grade in local recurrence and the disease-specific survival in chondrosarcomas. Bone Joint J 2018, 100-B (5), 662-666.
  3. Choi, B. B.; Jee, W. H.;  Sunwoo, H. J.;  Cho, J. H.;  Kim, J. Y.;  Chun, K. A.;  Hong, S. J.;  Chung, H. W.;  Sung, M. S.;  Lee, Y. S.; Chung, Y. G., MR differentiation of low-grade chondrosarcoma from enchondroma. Clin Imaging 2013, 37 (3), 542-7.
  4. Douis, H.; Singh, L.; Saifuddin, A., MRI differentiation of low-grade from high-grade appendicular chondrosarcoma. Eur Radiol 2014, 24 (1), 232-40.
  5. Crim, J.; Schmidt, R.;  Layfield, L.;  Hanrahan, C.; Manaster, B. J., Can imaging criteria distinguish enchondroma from grade 1 chondrosarcoma? Eur J Radiol 2015, 84 (11), 2222-30.
  6. Lisson, C. S.; Lisson, C. G.;  Flosdorf, K.;  Mayer-Steinacker, R.;  Schultheiss, M.;  von Baer, A.;  Barth, T. F. E.;  Beer, A. J.;  Baumhauer, M.;  Meier, R.;  Beer, M.; Schmidt, S. A., Diagnostic value of MRI-based 3D texture analysis for tissue characterisation and discrimination of low-grade chondrosarcoma from enchondroma: a pilot study. Eur Radiol 2018, 28 (2), 468-477.

3) Lines 59-61: the authors should add at least a reference.

Response:

Thank you for your pointing this out.

The following references were added to Lines 59-61.

Murphey, M. D.;  Flemming, D. J.;  Boyea, S. R.;  Bojescul, J. A.;  Sweet, D. E.; Temple, H. T., Enchondroma versus chondrosarcoma in the appendicular skeleton: differentiating features. RadioGraphics 1998, 18 (5), 1213-1237.

Higuchi, T.;  Taki, J.;  Sumiya, H.;  Kinuya, S.;  Nakajima, K.;  Namura, M.; Tonami, N., Characterization of cartilaginous tumors with 201Tl scintigraphy. Ann Nucl Med 2005, 19 (2), 95-9.

Kaya, G. C.;  Demir, Y.;  Ozkal, S.;  Sengoz, T.;  Manisali, M.;  Baran, O.;  Koc, M.;  Tuna, B.;  Ozaksoy, D.; Havitcioglu, H., Tumor grade-related thallium-201 uptake in chondrosarcomas. Ann Nucl Med 2010, 24 (4), 279-86.

4) Table 1 is confusing. In the paragraph “2.1. Development cohort”, the authors describe the 81 patients enrolled citing table 1. However, the reader is confused to see also “a validation cohort” that it is not describe in this part of the manuscript. The authors should consider to modify the table or move it at the beginning of the results.

Response:

Thank you for your suggestion.

Table 1 was moved to the beginning of the Results section.

5) Table 4 should be placed before figure 1.

Response:

Thank you for your suggestion.

In revised manuscript, Table 4 was placed before Figure 1.

6) Legends of Figures 3,4 and 5 should be improved. The authors should add letters on the figures describing x-ray, RMI, CT and histological images. Magnification/scale bar of histological images is missing. 

Response:

Thank you for your suggestion.

The figure legends were revised as follows:

Figure 3. Case 1. A 58-year-old woman presenting with shoulder pain. (a) Radiograph showed no periosteal reaction in the proximal humerus. (b) There was no endosteal scalloping or cortidal defect on CT. (c) MRI showed no extraskeletal mass, multilobular lesion, or abnormal signal in adjacent tissue. (d) Bone scan demonstrates uptake of 99mTc-MDP in the proximal humerus, but the uptake was not greater than uptake in anterior iliac crest. (e) 201Tl scan showed no increased uptake in the proximal humerus. Based on the results, overall radiological score rate was calculated as follows:

[0 points (X-ray) + 0 points (CT) + 0 points (MRI) + 0 points (bone scan) + 0 points (201Tl scan)] / 16 points (full marks) × 100 = 0%

(f) Microscopic image showed hypocellular tumor cells with an abundance of hyaline cartilage matrix. No cytological atypia or mitosis was observed. The final histological diagnosis was enchondroma.

Figure 4. Case 2. A 48-year-old woman presenting shoulder pain. (a) Radiograph showed no periosteal reaction. (b) CT showed endosteal scalloping and cortical defect. (c) MRI showed multilobular lesion, whereas no extraskeletal mass and abnormal signal in adjacent tissue was seen. (d) Bone scan showed increased uptake greater than anterior iliac crest. (e) 201Tl scan showed no increased uptake in the proximal humerus. Based on the findings, the overall radiological score rate was calculated as follows:

[0 points (X-ray) + 3 points (CT) + 3 points (MRI) + 2 points (bone scan) + 0 points (201Tl scan)] / 16 points (full marks) = 50.0%

(f) Microscopic image showed moderate cellularity, with cells embedded in hyaline matrix. There are closed chromatin pattern and inconspicuous nucleoli. The final histological diagnosis was grade I chondrosarcoma.

Figure 5. Case 3. A 58-year-old man showed a periosteal reaction on X-ray (a) ; endosteal scalloping and cortical defect on CT (b); and extraskeletal mass, multilobular mass, and abnormal signal intensity in adjacent soft tissue and bone marrow on MRI (c). Increased uptake greated than anterior iliac crest on bone scan and 201Tl scan were also seen (d,e). Based on the findings, the overall radiological score rate was calculated as follows:

[1 point (X-ray) + 3 points (CT) + 8 points (MRI) + 2 points (bone scan) + 2 points (201Tl scan)] / 16 points (full marks) = 100%

(f, g) Microscopic image showed bimorphic appearance of grade 1–2 chondrosarcoma and fibrosarcoma components. The final histological diagnosis of the tumor was dedifferentiated chondrosarcoma.

Scale bars were added to the histological images.

7) The authors should better organize the order of figures/tables following the text of the manuscript. Figures and tables should be placed near where they are cited. For example, table 5 should be moved after figure 5.

Response:

Thank you for your suggestion.

In the revised manuscript, the order of figures and tables were organized and these were placed near where they are cited.

Table 5 was moved after Figure 5.

Minor comments: It is not clear to me why there are some words in red throughout the manuscript.

Response:

Thank you for your pointing this out. The manuscript was reviewed and revised according to previous reviewers. To let the reviewers to easily find the points which were revised, the revised points were marked in red. 

Reviewer 4 Report

Interesting study on a classification system to differentiate chondroma from sarcoma.

Summary: please consider organizing the section starting from aims, methods, results and conclusion. First sentence of summary is awkward in this position. Please add a conclusion.

Abstract: please provide more details on the statistics and provide significant value.

Introduction: please justify every statements with a reference.

Materials and methods: please provide more details on inclusion/exclusion criteria of the 2 cohorts in a same chapter. How the patients were selected for one instead the other?

Results: interesting. Please add significant values in the abstract.

Discussion: please summarize and consider to delete 3rd and 4th paragraphs. Please avoid the use of first person (we, our).

Author Response

Reviewer 4. Comments and Suggestions for Authors

Interesting study on a classification system to differentiate chondroma from sarcoma.

Summary: please consider organizing the section starting from aims, methods, results and conclusion. First sentence of summary is awkward in this position. Please add a conclusion.

Response:

Thank you for your suggestion.

In the Summary section, background, methods, results, and conclusion were used to organize the section.

The first sentence was deleted from the summary, and the following sentences were added to the Summary.

Background: It is challenging to differentiate between enchondromas and atypical cartilaginous tumors (ACT)/chondrosarcomas.

Conclusion: Comprehensive assessment combining radiological findings is recommended to differentiate between enchondromas and ACT/chondrosarcomas.

Abstract: please provide more details on the statistics and provide significant value.

Response:

Thank you for your suggestion.

The details on the statistics were added to the Methods section, and the significant values were added to the Results section.

Introduction: please justify every statements with a reference.

Response:

Thank you for your suggestion.

In revised manuscript, every statement was justified with references.

Materials and methods: please provide more details on inclusion/exclusion criteria of the 2 cohorts in a same chapter. How the patients were selected for one instead the other?

Response:

Thank you for your suggestion.

This study was a retrospective study, and it is difficult to randomly divide patients into development cohort and validation cohort. So, the patients were assigned to development cohort and validation cohort according to the time when the tumors were diagnosed. About 80% of patients were included in the development cohort because large number of study patients is needed to develop to a scoring system with high reliability.

Details on inclusion/exclusion criteria of the two cohorts were added to the Methods section as follows:

Development cohort

This retrospective study included 98 patients with central cartilaginous tumors who underwent biopsy and/or tumor excision at the Department of Orthopedic Surgery of our hospital between October 2003 and December 2020 in our hospital. The patients were divided into development cohort and validation cohort according to the time of histological diagnoses. Among the patients, 81 patients with central cartilaginous tumors diagnosed between January 2008 and December 2020 were included in development cohort. Inclusion criteria were as follows: central cartilaginous tumors proven pathologically by specimens obtained from biopsy and/or surgery.

Exclusion criteria were as follows: recurrent tumors, metastatic tumors, and tumors without histological diagnoses.

Validation cohort

Seventeen patients with central cartilaginous tumors diagnosed between October 2003 and December 2007 were included in validation cohort. In validation cohort, the inclusion and exclusion criteria were the same as those for the development cohort.

Results: interesting. Please add significant values in the abstract.

Response:

Thank you for your comments.

Significant values were added to the abstract as follows:

Periosteal reaction on X-ray (P = 0.025), endosteal scalloping (P = 0.010) and cortical defect (P = 0.002) on CT, extraskeletal mass (P < 0.001), multilobular lesion (P < 0.001), abnormal signal in adjacent tissue (P = 0.004) on MRI, and increased uptake in bone scan (P = 0.002) and thallium scan (P = 0.027) was significantly correlated with final diagnoses.

In another cohort of 17 patients, the sensitivity, specificity, and accuracy of the radiological score rates for differentiation between enchondromas and ACT/chondrosarcomas were 88%, 89%, and 88%, respectively (P = 0.003).

Discussion: please summarize and consider to delete 3rd and 4th paragraphs. Please avoid the use of first person (we, our).

Response:

Thank you for your suggestion.

The following sentences were deleted from the Discussion section.

In the present study, extraskeletal mass, multilobular mass, and abnormal signal in the adjacent bone marrow and soft tissue on MRI were significantly correlated with final diagnoses. In general, enhanced MRI is useful for differentiating between benign and malignant bone tumors. However, it was reported that enhanced MRI is not useful for differentiating between enchondromas and ACT/grade I chondrosarcomas13. Choi et al. investigated the utility of MRI findings for differentiating between enchondromas and low-grade chondrosarcomas12. In the study, MRI findings, including tumor location, calcification, endosteal scalloping, cortical destruction, cortical ballooning, periosteal reaction, and abnormal signal intensity in the adjacent bone marrow and/or soft tissue, and histological diagnoses were evaluated. Among the findings, cortical destruction, soft tissue mass, and abnormal signal intensity in the adjacent bone marrow and soft tissue had a specificity of 100%. However, the intermediate signal on T1-weighted images and multilobular lesion on enhanced MRI had a high sensitivity for differentiation between enchondromas and low-grade chondrosarcomas.

This study showed that increased uptake of radionuclides in bone scans and 201Tl scans, were significantly correlated with final diagnoses. Although both enchondromas and chondrosarcomas showed increased uptake of radionuclides in bone scans, the level of uptake was different between the tumors30. To distinguish enchondromas and chondrosarcomas, a comparison of the radionuclide uptake level between the lesion and the anterior iliac crest was recommended8, 30. Murphey et al. reported that 82% of chondrosarcomas had radionuclide uptake higher than that of the anterior iliac crest. Furthermore, 79% of enchondromas had radionuclide uptake lower than that of the anterior iliac crest8. In previous studies, the correlation between 201Tl uptake and histological grade of cartilaginous tumors has been investigated9, 10. In a study on the correlation between 201Tl uptake and histological grade in 21 patients with cartilaginous tumors, increased uptake of 201Tl were observed in 0 of 3 (0%) patients with enchondromas, 0 of 9 (0%) grade I, 1 of 5 (20%) grade II, 1 of 1 (100%) grade III, 1 of 1 (100%) dedifferentiated, and 3 of 3 (100%) mesenchymal chondrosarcomas9. In another study of 201Tl scans in patients with cartilaginous tumors, increased 201Tl uptake was observed in 4 of 4 (100%) patients with grade III, 2 of 5 (40%) grade II, and 0 of 7 (0%) grade I chondrosarcomas10. Based on the previous reports and the present study, the 201Tl scan is useful in differentiating between low-grade and high-grade chondrosarcomas. However, it is not useful in differentiating between enchondromas and low-grade chondrosarcomas. Although positron emission tomography (PET) using 18fluorine-labelled fluorodeoxyglucose (18F-FDG) has contributed to the differentiation between benign and malignant tumors, cartilaginous tumors have lower 18F-FDG uptake compared to other types of malignancies32 33. In a systematic review of cartilaginous tumors, the mean maximum standardized uptake values (SUVmax) were 1.6 in benign cartilaginous tumor, 2.5 in grade I chondrosarcoma, and 6.0 in grade II/III chondrosarcoma26. In another study of PET-CT for differentiation between chondroma and chondrosarcoma, the sensitivity, specificity, and accuracy of PET-CT with a cutoff value of 2.2, were 95%, 94%, and 94%, respectively34. Based on the previous reports, PET can be included in the comprehensive assessment for the prediction of histological grade in cartilaginous tumors.

The terms of first person (we, our) were deleted from the Discussion section.

Round 2

Reviewer 4 Report

The authors adequately addressed the suggestions